# Natural Killer Cells Do Not Attenuate a Mouse-Adapted SARS-CoV-2-Induced Disease in *Rag2*^−/−^ Mice

**DOI:** 10.3390/v16040611

**Published:** 2024-04-15

**Authors:** Calder R Ellsworth, Chenxiao Wang, Alexis R Katz, Zheng Chen, Mohammad Islamuddin, Haoran Yang, Sarah E Scheuermann, Kelly A Goff, Nicholas J Maness, Robert V Blair, Jay K Kolls, Xuebin Qin

**Affiliations:** 1Division of Comparative Pathology, Tulane National Primate Research Center, Health Sciences Campus, 18703 Three Rivers Road, Covington, LA 70433, USA; cellsworth@tulane.edu (C.R.E.); cwang27@tulane.edu (C.W.); chenzhengsx@gmail.com (Z.C.); mislamuddin@tulane.edu (M.I.); sspence3@tulane.edu (S.E.S.); kgoff@tulane.edu (K.A.G.); nmaness@tulane.edu (N.J.M.); rblair3@tulane.edu (R.V.B.); 2Department of Microbiology and Immunology, Tulane University School of Medicine, New Orleans, LA 70112, USA; 3Departments of Medicine and Pediatrics, Center for Translational Research in Infection and Inflammation, Tulane University School of Medicine, New Orleans, LA 70112, USA; akatz15@tulane.edu (A.R.K.); hyang11@tulane.edu (H.Y.); jkolls1@tulane.edu (J.K.K.); 4Department of Pulmonary Critical Care and Environmental Medicine, Tulane University School of Medicine, New Orleans, LA 70112, USA

**Keywords:** SARS-CoV-2, MA30, COVID-19, T cells, B cells, NK cells

## Abstract

This study investigates the roles of T, B, and Natural Killer (NK) cells in the pathogenesis of severe COVID-19, utilizing mouse-adapted SARS-CoV-2-MA30 (MA30). To evaluate this MA30 mouse model, we characterized MA30-infected C57BL/6 mice (B6) and compared them with SARS-CoV-2-WA1 (an original SARS-CoV-2 strain) infected K18-human ACE2 (*K18-hACE2*) mice. We found that the infected B6 mice developed severe peribronchial inflammation and rapid severe pulmonary edema, but less lung interstitial inflammation than the infected *K18-hACE2* mice. These pathological findings recapitulate some pathological changes seen in severe COVID-19 patients. Using this MA30-infected mouse model, we further demonstrate that T and/or B cells are essential in mounting an effective immune response against SARS-CoV-2. This was evident as *Rag2*^−/−^ showed heightened vulnerability to infection and inhibited viral clearance. Conversely, the depletion of NK cells did not significantly alter the disease course in *Rag2*^−/−^ mice, underscoring the minimal role of NK cells in the acute phase of MA30-induced disease. Together, our results indicate that T and/or B cells, but not NK cells, mitigate MA30-induced disease in mice and the infected mouse model can be used for dissecting the pathogenesis and immunology of severe COVID-19.

## 1. Introduction

The COVID-19 pandemic, caused by coronavirus 2 (SARS-CoV-2), has profoundly impacted public health, with millions of cases and fatalities worldwide. The development of vaccines has contributed to a decline in the pandemic’s impact by reducing the spread of COVID-19, reducing the number of hospitalizations, as well as deaths associated with the disease [1,2,3,4]. However, new SARS-CoV-2 variants continue to arise, leading to an increased risk of breakthrough infection and subsequent hospitalizations in vaccinated people with comorbidities and immunocompromised conditions [5,6,7,8,9]. Although the World Health Organization (WHO) declared the COVID-19 pandemic over as a global health emergency on 8 May 2023, it remains a global health threat. The fight against COVID-19 is still a challenge. There is still a strong need to elucidate the mechanisms underlying the severe disease and death associated with COVID-19. A better understanding of these will not only help us develop specific treatment and vaccine strategies against the current threat raised by variants of concern but will also prepare us for combating potential outbreaks caused by the new variants [10].

It is well recognized that following SARS-CoV-2 infection, lymphocytes, including T cells, B cells, and NK cells, are activated, significantly contributing to the development of adaptive immunity against the infection and aiding in the clearance of the virus [11,12]. This is a key aspect of the immune response determining the clinical outcome after infection [13]. However, it remains unclear whether they have any roles in innate immunity against SARS-CoV-2. The rapid T cell responses against SARS-CoV-2 correlate with protection, but can be impaired in severe disease and are related to massive activation and lymphopenia [13]. The robust B cells and T follicular helper cell responses stimulated by COVID-19 vaccination contribute to the production of antibodies, which provide remarkable protection against SARS-CoV-2 infection, and reduction in hospitalization and death [13,14]. Notably, the depletion of B cells with anti-B antibodies in Non-Hodgkin lymphoma and chronic lymphocytic leukemia patients has been shown to weaken the humoral response and the production of Spike-specific antibodies following vaccination, increasing the risk of breakthrough infections [15,16,17]. Effector T and B cell responses following infection or vaccination also contribute to the subsequent development of specific immunological memory [18]. Clinical studies show that NK cells have heightened activation though are functionally impaired within the COVID-19 immune response. Bronchoalveolar lavage and peripheral NK cell analyses revealed the increased cytokine production and expression of cytotoxic proteins (granzyme B and perforin) in both mild and severe patient cohorts in comparison with SARS-CoV-2 immunoglobulin G (IgG) seronegative healthy controls [19]. However, ex vivo studies demonstrated the reduced functional activity of peripheral blood NK cells in targeting infected cells due to the impaired activation of NK activating receptors (NKp30, NKp44, and NKp46), increased inhibitory receptors, and target cell evasion via the decreased expression of ligands for the NK activating receptor, NKG2D [20,21]. Together, this emerging evidence demonstrates the critical roles of T, B, and NK cells in adaptive immunity protecting against SARS-CoV-2 infection. However, the exact roles that each of these cells plays in primary immunity when restricting SARS-CoV-2, at the acute phase of infection, have not been experimentally investigated.

In this study, we aimed to investigate the function and contribution of T, B, and NK cells in the context of primary SARS-CoV-2 infection using a SARS-CoV-2-MA30 (MA30) model, a mouse-adapted strain that was recently generated [22]. To explore the role of T and B cells in COVID-19, we utilized *Rag2*^−/−^ mice in a C57BL/6J (B6) background. *Rag2*^−/−^ mice are deficient in the development of mature T and B cells due to the deficiency of *RAG2*, a gene essential for V(D)J recombination and lymphocytic development [23]. We also employed a depleting anti-NK cell antibody in *Rag2*^−/−^ mice to investigate the function of NK cells in this context. We demonstrated that the inoculation of B6 mice with MA30 results in a severe phenotype, recapitulating some key features of human severe COVID-19. We compared the infection patterns of MA30 in B6 mice to the well-established human transgenic *K18-hACE2* mouse model infected by CoV-2-WA1 strains USA-WA-1/2020 (WA1) [24,25], highlighting the differences in viral tropism and cytokine responses between the two strains. Additionally, *Rag2*^−/−^ mice exhibited increased susceptibility to viral infection and prolonged weight loss, confirming the importance of T and/or B cells in the immune response against SARS-CoV-2, as the depletion of NK cells did not significantly impact the course of the disease in *Rag2*^−/−^ mice. Together, our results indicate that T and B cells, but not NK cells, attenuate MA30-induced disease in mice.

## 2. Methods

### 2.1. Ethical Compliance and Animal Models

Our research adhered to the ethical guidelines approved by the Institutional Animal Care and Use Committees at Tulane University. The study involved the use of several mouse models, namely *K18-hACE2*^+/−^ in a B6 background (Stock No. 034860), B6 mice (Stock No. 000664), and *Rag2*^−/−^ in a B6 background (Stock No. 008449), all procured from the Jackson Laboratory (Bar Harbor, ME, USA) and maintained at Tulane University’s animal facility.

### 2.2. Inoculation with SARS-CoV-2

We utilized an ancestral SARS-CoV-2 strain, specifically the USA-WA1/2020 isolate (NR-52281), sourced from the CDC through BEI Resources, NIAID, NIH (Manassas, VA, USA). Virus was expanded on VeroE6/TMPRSS2 cells using DMEM media with 2% FBS, followed by harvesting and sequencing to confirm homogeneity with the original isolate. The CoV-2 MA30 variant was acquired directly from Dr. Perlman as a seed stock and expanded and characterized at Tulane using the same methodology as above to generate a stock for in vivo infections. In our ABSL3 facility, mice were administered intranasal infections of 50 μL total volume per nostril with SARS-CoV-2 at varying doses depending on the experimental requirement.

### 2.3. NK Cell Depletion

A total of 250 μg of anti-NK1.1 antibody (BE0036, Bio X Cell, Lebanon, NH, USA) was diluted in PBS and administered via sterile intraperitoneal injection. As an appropriate control, we employed an age-matched cohort of *Rag2*-deficient mice treated with IgG2a (2A3, Bio X Cell, BE0089), which shares the F1 region with the NK1.1 antibody.

### 2.4. Histopathological Examination

For histological studies, lung tissue sections were stained using the standard hematoxylin and eosin (H&E) method and digitally imaged with Nanozoomer S360 scanner. A veterinary pathologist classified areas of edema and cellular infiltration using Halo Indica Labs software, then trained a neural network to identify these regions autonomously, with automatic annotations later reviewed for accuracy.

### 2.5. Detection of Viral RNA

Tissue samples for RNA were stored in Trizol reagent (15596026; Invitrogen, Carlsbad, CA, USA) and RNA quantification was performed using the Thermo Scientific NanoDrop 2000 Spectrophotometer (Wilmington, DE, USA). Using 100 ng of total RNA, the detection of subgenomic N viral RNA was conducted with Taqpath 1-Step Multiplex Master Mix (A15299; Thermo Fisher, Wilmington, DE, USA) and specific FAM-labeled primers. Quantification was achieved through standard Cq values, using an ABI QuantStudio 6 system (Waltham, MA, USA 02451).

### 2.6. Immunohistochemical Staining

Lung tissue sections were fixed in zinc formalin, embedded in paraffin, and then subjected to epitope retrieval using solutions of varying pH (Vector Labs H-3301 and H-3300) as previously described in [26,27]. Blocking was performed using BSA/or serum, followed by incubation with primary and secondary antibodies. The dilutions and the primary antibodies were 1:1000 anti-SARS (NR-10361, BEI, Manassas, VA, USA), 1:500 anti-Spc (WRAB-9337, Seven Hills Bioreagents, Cincinnati, OH, USA), and 1:20 anti-cytokeratin antibody (Z0622, Dako, Glostrup, Denmark). The corresponding secondary antibodies were added as detailed in [27]. Digital imaging was executed using a Zeiss Axio scan. Z1. (Jena, Germany).

### 2.7. Illumina Total RNA Library Prep, Sequencing, and Analysis

Prior to Illumina total RNA library construction, DNase-treated total RNA was quantitated using the Qubit RNA HS assay kit (Thermo Fisher Scientific, #Q32855, Waltham, MA, USA 02451). RNA quality (RNA Integrity Number, RIN) was determined on an Agilent TapeStation 4150 using an Agilent RNA ScreenTape (Agilent, #5067–5576, Santa Clara, CA, USA). A total of 0.3 ug of each total RNA was applied to generate total RNA libraries using the Illumina TruSeq Stranded total RNA Sample Preparation kit following the Illumina TruSeq Stranded total RNA Sample Preparation Guide generated by IIIumina (Illumina, San Diego, CA, USA).

Final cDNA libraries containing TruSeq RNA CD indexes (Illumina, 20019792) were quantitated using the Qubit dsDNA HS assay kit (Thermo Fisher Scientific, Q32854). The quality of the libraries was determined by running each on an Agilent TapeStation 4150 using an Agilent D1000 ScreenTape (Agilent: 5067-5582, Santa Clara, CA, USA). Smear analysis was performed using Agilent TapeStation Software (Version 4.1.1) with a range of 200–600 bp to determine the average size of each library. Size and concentration were then used to calculate the molarity of each library.

All libraries were pooled at a final concentration of 750 pM with a spike-in of 2% PhiX control library v3 (Illumina, FC-110-3001, Jena, Germany). A mixture of pooled libraries was loaded on an Illumina NextSeq P3 (200) reagent cartridge (Illumina, 20040560, Jena, Germany). Paired-end and dual indexing sequence, 100 × 8 × 8 × 100, was performed on the NextSeq2000, yielding approximately 47 M paired-end reads per sample. Fastqs generated by Illumina BaseSpace DRAGEN Analysis Software (Version 1.2.1) were applied for further data analyses. Fastqs were uploaded to the Banana Slug Analytics platform for downstream analysis (Slug Genomics, UV Santa Cruz, Agilent, Santa Clara, CA, USA).

Cuffdiff and EdgeR were conducted to determine the differentially expressed genes (DEGs) (*p* < 0.05). Gene enrichment analysis was conducted through ShinyGo 0.80 using DEGs generated by EdgeR as described in [28]. The volcano plot was generated by Graphpad Prism. All curated datasets were deposited in the Sequence Read Archive pending a BioProject number.

The RNA-seq data was deposited at GEO with accession number: GSE263657.

### 2.8. Preparation of Single Cells from Mouse and Flow Cytometry

We followed our established protocol to prepare single cells from mouse lungs as detailed in [29]. Briefly, mice were euthanized by using CO_2_ in a BSL-3 facility and peripheral blood was collected in an EDTA-tube. The lungs were perfused with PBS, harvested, and processed for single cell preparation. The lungs were mechanically dissociated and incubated for 30 min in 10 mL of 1 × HBSS containing 0.75 mg/mL collagenase IV (Worthington, Lakewood, NJ, USA) and 20 μg/mL DNaseI (Worthington) at 37 °C. After digestion, the cell suspension was passed through 40 μm cell strainers and maintained on ice. The suspension was washed twice with PBS (500× *g*, 8 min). PBMC red blood cells were lysed using ACK-Lysis buffer (Gibco) for 10 min at room temperature and washed twice with cold PBS. A single cell suspension from the lungs and PBMC cells was fixed with 2% PFA for 30 min on ice, washed twice with cold PBS, and finally resuspended in FACS buffer (PBS containing 2% FBS), before flow cytometry staining.

The single cell suspension from the lungs and PBMC cells was incubated with 1:200 anti-CD16/32- FcγRIII/II (Clone 93, Cat# 48-0161-80, ebioscience) for 15 min to block non-specific Fc receptor binding. Aqua live and dead dye (Invitrogen, Cat # L34957A) was used to distinguished live/dead cells. For phenotyping of NK cells, T cells and B cells, we used the following pre-conjugated antibodies: CD45-e450 (Clone 30-F11, ebioscience Cat# 48-0451-82), NK1.1-FITC (Clone PK136, ebioscience Cat# 11-5941-85), CD3-APC (Clone 17A2, Invitrogen Cat# 17-0032-82), CD19-PE (Clone 1D3, Invitrogen Cat# 12-0193-82), and CD11b-PE-cy7 (Clone M1/70, Invitrogen Cat# 25-0112-82). In general, a 1:100 dilution of the above antibodies was used for the staining. The antibody cocktails were added and incubated at 4 °C in the dark for 30 min. After washing twice with FACS buffer, the samples were acquired on BDLSR Fortessa and analyzed using FACS Diva v.6.1.3 software (BD: https://www.bdbiosciences.com/en-ca/products/instruments/software-informatics/instrument-software/bd-facsdiva-software-v-6-1-3.643629 (accessed on 24 January 2024)).

### 2.9. Statistics

The data representation is in the form of mean ± SEM. For multiple group comparisons over time, a two-way ANOVA with subsequent Sidak’s post hoc tests was applied. For two-group or multi-group comparisons, we used the unpaired Student’s *t*-test or one-way ANOVA with Tukey’s multiple comparisons, considering *p* < 0.05 as statistically significant.

## 3. Results

### 3.1. MA30-Infected B6 Mice Recapitulate the COVID-19 Phenotype

We characterized the phenotype of acute disease in 8-week and 12-week-old wild-type B6 males infected with SARS-CoV-2(MA30) (Figure 1A–E), a mouse-adapted strain previously established by Dr. Pelman’s lab [22]. They reported MA30 as highly virulent such that 5000 PFUs caused a lethal disease in young (6–10 week) BALB/c mice and aged (24 weeks old) B6 mice, but not in 6–10-week-old B6 mice [22]. A younger cohort of B6 mice (8 weeks old) were infected with a moderate or low dose (moderate dose: 5 × 10^4^, and low dose: 1 × 10^4^ TCID_50_), and all mice survived infection after a monitoring period of 15 days (Figure 1A). The moderate-dose infected B6 mice lost a greater percentage of body weight than the low-dose B6 infected mice (Figure 1A). The amount of subgenomic N viral RNA in the lungs of these mice was similar at 15 days post-infection (DPI) (Figure 1B). To assess the dose-dependent effect of the virus, 12-week-old B6 mice were inoculated with three different doses of MA30 (high dose: 2 × 10^5^, moderate dose: 5 × 10^4^, and low dose: 1 × 10^4^ TCID_50_) and monitored for body weight (Figure 1C) and survival (Figure 1D). We noticed that a high dose of MA30 resulted in poor health and various stress behaviors in mice––such as nasal congestion and ocular discharge––highlighting the sensitivity of the mice to the effects of the virus. The results showed that increasing the dose positively correlated with accelerated weight loss, and higher lethality at 4-to-6 DPI in the infected mice (Figure 1C,D). All mice that received the high dose died within 7 days, which precluded a comparison of body weight changes between medium and lose-dose recipients at later time points (Figure 1D). We used q-RT-PCR to measure the viral nucleoprotein (N)-subgenomic viral RNA in the lungs of all these mice, a well-recognized method to determine replicating virus [30], and we noticed a dynamic trend for diminished viral subgenomic N RNA at lengthened post-infection dates (3–10 DPI) regardless of the infection dose (Figure 1E). Our results also showed a significant age-dependent effect on these mice using either the moderate or low dose of SARS-CoV-2 (Figure 1F). Altogether, we showed that MA30 can successfully infect young B6 mice in a dose-dependent manner.

### 3.2. Comparison of MA30 vs. the K18-hACE2 SARS-CoV-2-WA1 Model

We investigated the infection pattern of MA30 in B6 mice and compared them with the well-established human transgenic *K18-hACE2* mouse model exposed to the original CoV-2-WA1 strain, USA-WA1/2020. SARS-CoV-2-infected *K18-hACE2* mice is a severe COVID-19 mouse model that has been widely used for testing therapeutic countermeasures and vaccines, as well as investigating the pathogenesis of severe COVID-19 [24,31,32,33,34]. We examined the nasal cavity and lung compartments at 3 DPI using a matched lethal dose (LD90) for MA30 in B6 mice (5 × 10^4^ TCID_50_, IN) and WA1-infected K18 (1 × 10^4^ TCID_50_, IN), respectively. In both models, we identified virus-positive cells within the nasal turbinates (Figure 2A,B). In the pulmonary compartment, the MA30 strain displayed a more pronounced presence within the epithelium of the conducting airways (bronchus and bronchioles), while the viral infection in the *K18-hACE2* mice was primarily localized within the epithelial cells of the alveolus (Figure 2C–E). However, both the MA30 and WA1 strains demonstrated infection of both the bronchial and alveolar epithelium (Appendix A). Notably, infection of the bronchial epithelium by the MA30 strain was associated with epithelial sloughing of infected and necrotic cells into the airway lumen (Appendix A). We found that many alveolar type 2 (AT2) cells were infected in the *K18-hACE2* mice as compared with the B6 mice (Appendix A). The finding in the infected *K18-hACE2* mice is consistent with the previous observation published in [35]. Furthermore, H/E staining revealed a tendency for increased cellular infiltrate in the *K18-hACE2* mouse model, whereas the level of cellular inflammation in the B6 mice infected with MA30 was less pronounced (Figure 2F–H). Together, our results document divergent pathological phenotypes between MA30 in B6 and CoV2-WA1 in *K18-hACE2* mice. WA1 in *K18-hACE2* mice primarily targeted alveolar epithelial cells and resulted in interstitial pneumonia, whereas MA30 in B6 primarily targeted the larger conducting airways resulting in bronchointerstitial pneumonia.

### 3.3. Deficiency of Rag2 Accelerates MA30-Induced Disease in Mice

To elucidate the role of T and/or B cellular immunity in COVID-19, we infected the aged (8-week-old-matched *Rag2*^−/−^) mice [36], as well as WT B6 mice with two different doses of MA30 (5 × 10^4^ and 1 × 10^4^ TCID_50_). To reveal the ongoing effects of the virus during the transition from the acute to chronic phase of infection, we monitored the infected mice up to 15 DPI. *Rag2*^−/−^ mice exhibited significantly greater weight loss compared with WT B6 mice. This effect was seen with both doses of the infection (Figure 3A,B). Unlike WT B6 controls, *Rag2*^−/−^ mice failed to fully recover their original body weight at either 8 DPI or 15 DPI (Figure 3A,B). qRT-PCR analysis was used to detect viral N subgenomic RNA in the multiple tissues of the mice at 15 DPI. Infected *Rag2*^−/−^ mice had a significant increase in the viral RNA in the lung but not in the brain, spleen, or kidney compared to WT mice (Figure 3C). This finding was further supported by immunohistochemical staining of viral S protein, in lung tissue, showing that the infected *Rag2*^−/−^ mice had a greater expression of S protein at 15 DPI than the infected WT B6 mice (Figure 3D,E and Appendix A). Histologically, we detected no differences in inflammation in MA30-infected *Rag2*^−/−^ and B6 mice at 15 DPI (Appendix A). Moreover, we detected the percentage of CD3+ positive T and CD19 positive B cells in infected B6 and *Rag2*^−/−^ mice at 3 DPI via flow cytometry analysis (Appendix A). As expected, the infected *Rag2*^−/−^ had a significantly lower percentage of T and B cells in both lung and blood than the infected B6 mice (Appendix A). Together, these findings suggest that the absence of matured T and/or B cells contributes to exacerbated weight loss and delayed viral clearance in *Rag2*^−/−^ mice. 

We further explored the underlying mechanism by comparing pulmonary transcriptomic changes of the infected *Rag2*^−/−^ with B6 mice at 3 DPI (Figure 4). Among 15 perturbed pathways, the pathways for primary immunodeficiency, B cell receptor signaling, T cell receptor signaling, Th17 cell differentiation, and Th1 and Th2 cell differentiation were significantly dysregulated in the infected *Rag2*^−/−^ mice as compared to the B6 mice (Figure 4A and Appendix A). Moreover, the infected *Rag2* deficient mice had significantly reduced transcript expression of *Cd8a*, *Cd22*, *Cd79b*, *Igj* (immunoglobulin joining chain gene), *Cd8a*, *H2-Ob* (MHC class II antigen O beta gene), and Btla (B and T lymphocyte attenuator) than the infected B6 mice (Figure 4B and Appendix A). *Cxcl9*, *Cxcl1. Ccl2*, *Ccl7*, *Cxcl2* and *Cxcl5* were significantly up-regulated in the lungs of the infected *Rag2*^−/−^ mice as compared to the infected B6 mice (Figure 4B).

### 3.4. Depletion of NK Cells in Rag2^−/−^ Mice Did Not Accelerate MA30-Induced Severe Disease

NK cells are derived from the lymphocyte lineage, primarily contribute to innate immunity, and do not rely on V(D)J recombination for maturation [37]. Previous studies have shown that patients at the early phase had increased percentages of NK cells in the circulation [38] and in BAL fluid [39] as compared with patients before the infection. Consistently, we have found that MA30-infected B6 mice had a higher percentage of NK cells in blood and lungs than naïve B6 mice. Moreover, the infected *Rag2*^−/−^ had a higher percentage of NK cells in blood and lungs than the infected B6 mice (Appendix A). To experimentally investigate whether NK cells play a protective role in the context of COVID-19 at the early phase of infection, we depleted NK cells in *Rag2*^−/−^ mice. We focused on these mice to isolate and highlight the contribution of NK cells in the absence of other lymphocytes. To specifically deplete NK cells, we treated 18-week-old *Rag2*^−/−^ mice both 2 h before and 4 days after infection with CoV2-MA30 (2 × 10^4^ TCID_50_) with an anti-NK1.1 antibody, a widely used antibody for depleting NK cells in the mouse model [40,41,42] or an IgG2a isotype control (Figure 5). Percentage body weight and survival rate comparisons revealed no significant differences between the NK-depleted mice and IgG2a-treated controls (Figure 5A,B). Moreover, the viral RNA in the NK-depleted group at the end of the study period were comparable to that of the control group (Figure 5C). Histological examination of lung sections showed no noticeable differences in pulmonary edema or cellular inflammation scores between the two groups (Figure 5D,E). Successful depletion of NK cells was confirmed in the lungs of our experimental animals through qRT-PCR analysis of *Klrb1c*, a marker for NK cell presence, as used previously [40,41,42] (Appendix A). These findings suggest that the depletion of NK cells does not significantly impact the course of MA30-induced disease in the setting of *Rag2* deficiency. 

## 4. Discussion

Our study explored the pathogenic profile of the MA30 strain, developed in Dr. Perlman’s laboratory through serial passages in immunocompromised mice [22]. Our analysis revealed that when younger B6 mice (8–12 weeks old) are inoculated with this strain, they exhibit some symptoms of severe COVID-19, including an associated weight loss, significant lung abnormalities, and increased mortality [43]. Notably, our study further demonstrates that aging mice from only 8 to 12 weeks significantly exacerbates these symptoms, highlighting the age-dependent severity of the disease. 

Comparing the MA30-infected B6 mice with traditional models using wild-type viral strains and humanized *K18-hACE2* mice, we observed distinct pathological phenotypes. The infected B6 mice exhibited epithelial cell infections in the entire airway––including the nasal cavity, bronchus, and bronchioles and alveoli––while the infected *K18-hACE2* mice exhibited epithelial cell infections in the nasal cavity and alveoli, with little to no infection in the bronchus and bronchioles. This viral tropism is comparable to the viral tropism observed from autopsy studies of deceased COVID patients shown in [44,45,46]. Of note, the previous study also documented that in the infected *K18-hACE2* mice, there was an infection of alveolar type I cells (AT1) in addition to AT2 infection, which is one of the contributing factors for the severity seen in the infected *K18-hACE2* mice [47]. However, whether MA30 also infects the AT1 cells in B6 mice remains unclear and requires further investigation. Further, we found that the infected B6 mice developed more severe peri-bronchial inflammation with similar pulmonary edema and less lung interstitial inflammation compared with the infected *K18-hACE2* mice. The pulmonary alveolar edema is one of major pathological changes seen in severe COVID-19 [44,48,49,50,51,52]. Together, the viral tropism and pathological changes seen in the infected B6 mice recapitulate several key histopathological features seen in severe COVID-19 patients. This underscores the value of the MA30 model in understanding the pathogenesis of severe COVID-19, especially its pulmonary complications. 

Our research further sheds light on the roles of T and B cells in the adaptive immune response during the late stages of infection, emphasizing their importance in viral clearance. We found that the absence of these cells leads to prolonged viral persistence and incomplete weight recovery. However, the relative contributions of CD4+ T cells, CD8+ T cells, or B cells to the attenuation of MA30-induced disease during the acute phase remain elusive and warrant further dissection with the elimination of these cells in the mice by using cell-targeted ablation tools [29,53,54,55,56]. 

Further, the infected *Rag2*^−/−^ mice had a significant dysregulation in the primary immunodeficiency, B cell receptor signaling pathway, T cell receptor signaling pathway, Th17 cell differentiation, and Th1 and Th2 cell differentiation. These pathways are closely related to T and B cell differentiation and activation, and antibody production [57,58]. Key genes related to these pathways (for example *Cd8a* for primary immunodeficiency and antibody production [59], *Cd22*, *Cd79b*, and *Igj* for the B cell receptor signaling pathway and antibody production [60,61,62], *Cd8a* for the T cell receptor signaling pathway and *H2-Ob* for Th17 cell differentiation [59], and *Btla* and *H2-Ob* for Th1 and Th2 cell differentiation and activation [63,64] were down-regulated in the infected *Rag2*-deficient mice. These defects and gene down-regulation can be attributed to the deficiency in Rag2 function in *Rag2*^−/−^ mice in the response to MA30 infection [57,58]. However, how the down-regulated genes and dysregulated pathways in *Rag2*^−/−^ mice attenuate the innate immunity against acute MA30 infection remains unclear. 

Additionally, we investigated the role of NK cells in MA30-induced disease using *Rag2*^−/−^ mice. Our finding suggests that the innate function of NK may not contribute to alleviating MA30-induced disease. Of note, a previous study indicated that there was hyperactivation of NK cells in *Rag2*^−/−^ mice [65]. Although NK cell depletion in B6 mice would provide further insight into the role of NK in COVID-19, we do not expect the outcome to change dramatically considering that this hyperactivation of NK cells in *Rag2*^−/−^ mice would have been suppressed by our cell ablation approach. In severe COVID patients, NK cells are exhausted due to the induced expression of the inhibitory molecules, including the overexpression of PD-1 on its surface [66,67]. Additionally, SARS-CoV-2 infection impairs NK cell functions by activating the LLT1-CD161 axis, leading to reduced cytotoxicity [68]. NK cells in COVID-19 patients show dysfunction, characterized by decreased quantity and impaired ability to reduce viral protein levels, similar to tumor-associated NK cells [69]. NK cells are exhausted and thereby play a negligible role in the acute phase of severe COVID-19 disease in patients [20,21,66,67,68,69]. Our results support those clinical findings though further investigation of the exhausted and activated status of NK cells is required. Further, our results do not rule out the potential role of NK cells in long-COVID scenarios. Our focus on the acute phase leaves open questions about the broader implications of NK cells, especially considering their known interactions with T and B cells in regulating the adaptive immune response. The minimal impact of NK cells in our acute model suggests possible compensatory mechanisms, such as those involving the increased interferon response, which warrant further investigation.

## Figures and Tables

**Figure 1 viruses-16-00611-f001:**
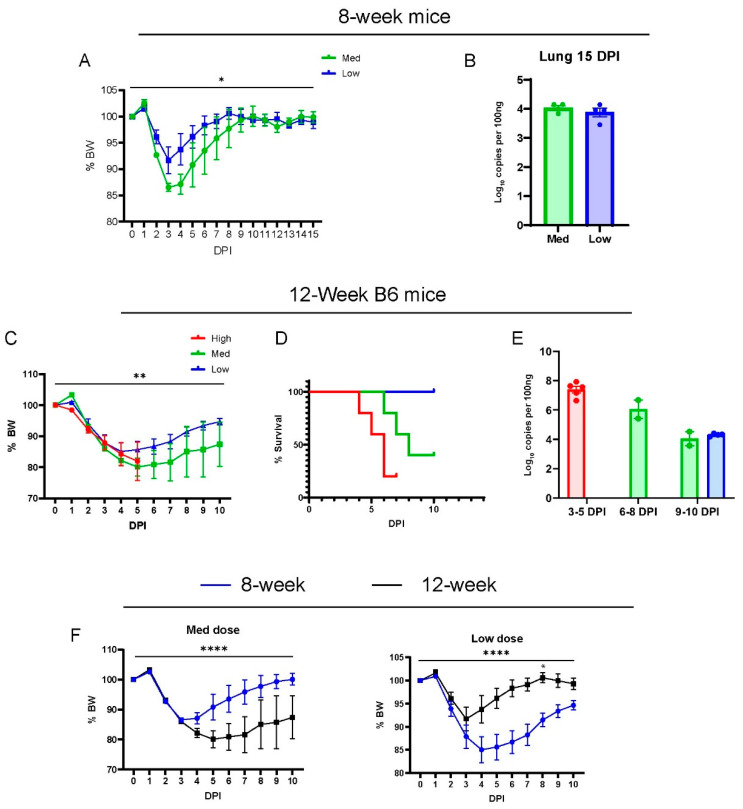
Dose-dependent effects of MA30 infection in B6 mice: (**A**) Body weight changes in 8-week-old mice infected with medium (Med) and low (Low) doses (*n* = 4 mice/group). *p* = 0.0248, comparing the body weight (BW) changes between the groups by two-way ANOVA analysis. (**B**) Viral subgenomic N RNA in the infected 8-week-old mice (from **A**) was measured using qPCR. (**C**) Body weight changes after the infection. ** *p* = 0.0051 (*n* = 5 per group) derived from comparing low- vs. medium-dose groups by mixed-effects analysis. (**D**) Survival rates over 10 days post infection (DPI) for each group (*n* = 5/group). The log-rank (Mantel-Cox) test showed a *p*-value of 0.0104. (**E**) Viral subgenomic N RNA was quantified through qPCR. The *y*-axis represents normalized values (log_10_ copies per 100 ng). Twelve-week-old B6 mice (shown in **C**–**E**) were divided into three groups and intranasally inoculated with three different doses of CoV2-MA30: high (2 × 10^5^ TCID_50_), medium (5 × 10^4^ TCID_50_) (Med), and low (1 × 10^4^ TCID_50_). Data from the high-dose group were derived from an independent experiment. (**F**) Body weight differences between the 8-week-old and 12-week-old mice infected with both medium and low doses of MA30. **** *p <* 0.0001 was analyzed by mixed-effects analysis for the left panel or two-way ANOVA analyses for the right panel (*n* = 4/group). * *p* < 0.05 was analyzed by using Sidak’s multiple comparisons.

**Figure 2 viruses-16-00611-f002:**
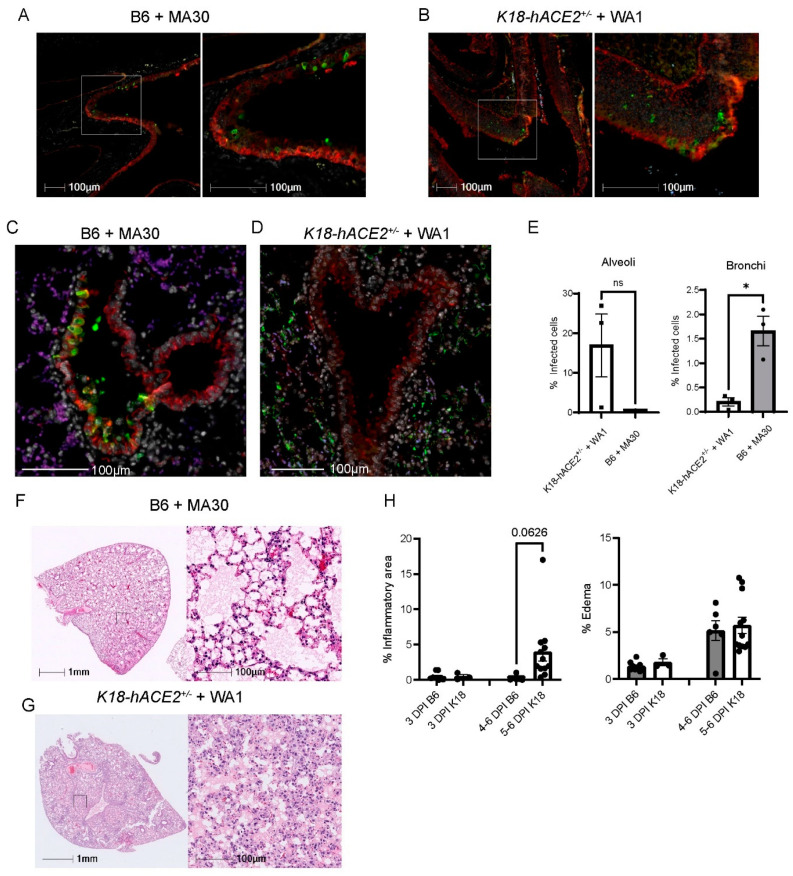
Viral tropisms and lung pathological changes in MA30 infected B6 and CoV2-infected *K18-hACE2^+/−^* mice. Twenty-week-old B6 and *K18-hACE2* mice shown in (**A**–**E**) were intranasally infected with CoV2 MA30 (5 × 10^4^ TCID_50_) and CoV2 MA1 (1 × 10^4^ TCID_50_), respectively. (**A**,**B**) Epithelial cell infection of the nasal turbinates in the infected B6 (**A**) and *K18-hACE2* mice (**B**) at 3 DPI. (**C**,**D**) Epithelial cell infection in the bronchus and bronchioles of CoV-2 MA30-infected B6 mice at 3 DPI (**C**) and in the alveoli’s epithelial cells of the WA1-infected *K18-hACE2* mice at DPI (**D**). Cross-sections of the nasal turbinate and lung from a 20-week-old B6 mouse and a *K18-hACE2* mouse. Immunostaining with the anti-cytokeratin marker in red and the anti-SARS-CoV-2 N protein in green. (**E**) Viral tropism was assessed by quantifying infected cells within the bronchial epithelium (**left panel, E**) and lung alveoli (**right panel, E**) of the infected mice (*n* = 3 mice per group). Viral infection was significant in the bronchial epithelium but not the alveoli, as indicated by unpaired *t*-test (*p* = 0.0100 and 0.1040, respectively). (**F**) Extensive edema in the lungs of infected B6 mice via hematoxylin and eosin (H/E) staining. (**G**) Extensive inflammation seen in the infected *K18-hACE2* mice. (**H**) Quantitative analysis of lung histological changes in the infected mice (*n* = 9 B6 at 3 DPI, *n* = 3 *K18-hACE2* at 3 DPI, *n* = 6 B6 at 5–6 DPI, *n* = 12 at 5–6 DPI). Mouse lung tissues shown in (**F**–**G**) were collected from multiple experiments with *K18-hACE2* male and female mice infected with 2.0 × 10^5^ PFU/mouse, and B6 mice infected with the high dose of 5 × 10^4^ TCID_50_. Pattern recognition algorithms were used to quantify the percentage of lung area affected by inflammation or edema. * *p* < 0.05.

**Figure 3 viruses-16-00611-f003:**
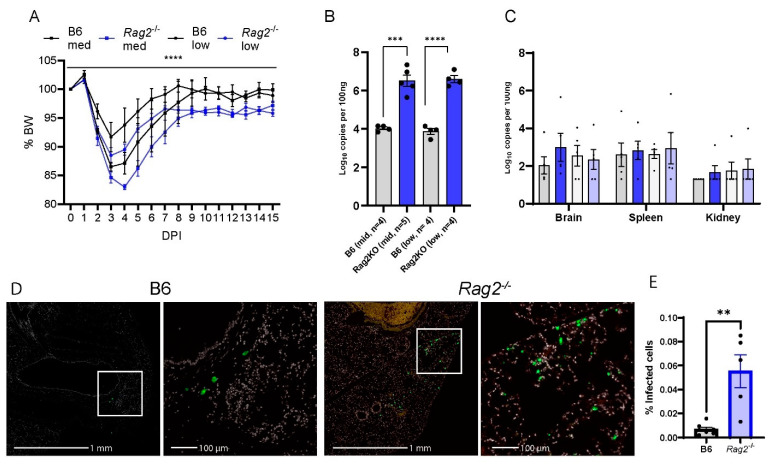
*Rag2* deficiency accelerates body weight loss and sustains viral load. B6 and *Rag2*^−/−^ 8-week-old mice were intranasally infected with a medium (Med) and low (Low) dose of CoV2-MA30 and monitored for 15 DPI (B6 Med, *n* = 4; *Rag2*^−/−^ med, *n* = 4; B6 Low, *n* = 5; *Rag2*^−/−^ low, *n* = 4). (**A**) Body weight changes in the infected *Rag2*^−/−^ and B6 mice. *p* < 0.0001, comparing BW combined by medium- and low-dose groups by three-way ANOVA. (**B**) Viral subgenomic RNA in the mice. Viral subgenomic N RNA was measured through qPCR, and significant difference was found between *Rag2*^−/−^ and B6 groups in both medium- (*p* = 0.0002) and low-dose (*p* < 0.0001) groups, using unpaired *t*-test. (**C**) Viral subgenomic N RNA detection in multiple extra-pulmonary tissues, including the brain, spleen, and kidney. (**D**,**E**) Viral infection detected via viral S protein staining. Anti-SARS S protein antibody in green. Representative images of viral S staining of B6 and *Rag2*^−/−^ mice infected with the medium dose of CoV2-MA30 at DPI 15. (**E**) Quantification of viral S protein positive cells in the lungs of all medium-dose and some low-dose mice with *p*-value 0.0019 from two-tailed unpaired *t*-test (*n* = 7 B6, *n* = 5 *Rag2*^−/−^). ** *p* < 0.01, *** *p* < 0.001, **** *p* < 0.0001.

**Figure 4 viruses-16-00611-f004:**
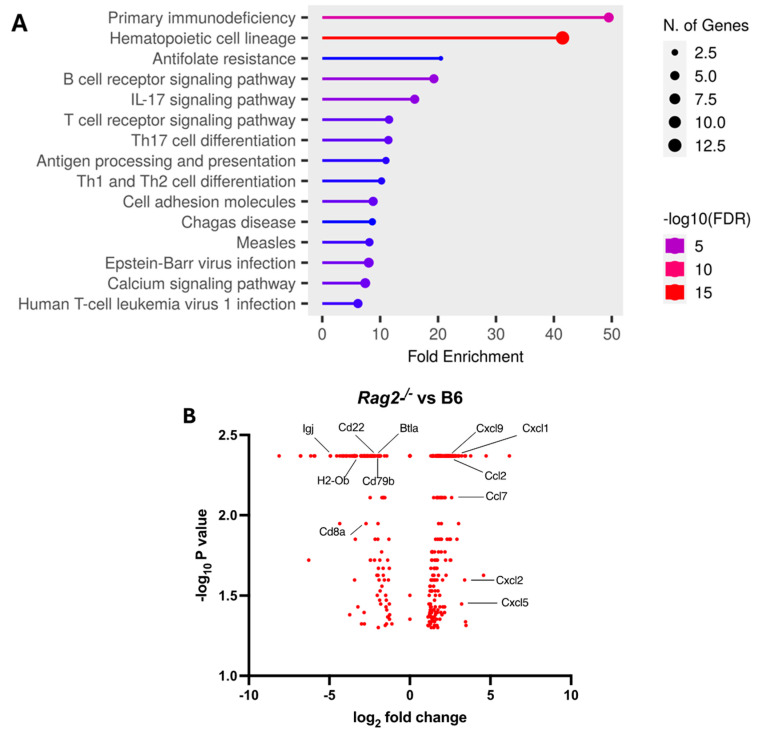
Transcriptomic analysis of the lungs of the infected *Rag2*^−/−^ and B6 (*Rag2*^+/+^) mice: Bulk RNA seq analyses were conducted on the lungs of the infected 14-week-old *Rag2*^−/−^ (*n* = 3) and B6 males (*n* = 3) at 3 days post-infection (DPI) by MA30 (5 × 10^4^ TCID_50_). Differentially expressed genes (DEGs) were generated by Cuffdiff and EdgeR (*p* < 0.05). (**A**) Dot plot of changed pathways in the lungs of *Rag2*^−/−^ mice as compared to B6 mice. EdgeR DEGs (Appendix A) were used for ShinyGo 0.80 gene enrichment analysis (need to cite: ShinyGO: a graphical gene-set enrichment tool for animals and plants). (**B**) Volcano plot showing fold-change and *p*-value of DEGs between *Rag2*^−/−^ mice and B6 mice generated by Cuffdiff (Appendix A). Key genes involved in primary immunodeficiency pathway, B cell and T cell receptor signaling pathways, Th1, Th2, and Th17 cell differentiation pathways, and interferon response genes are indicated with black line.

**Figure 5 viruses-16-00611-f005:**
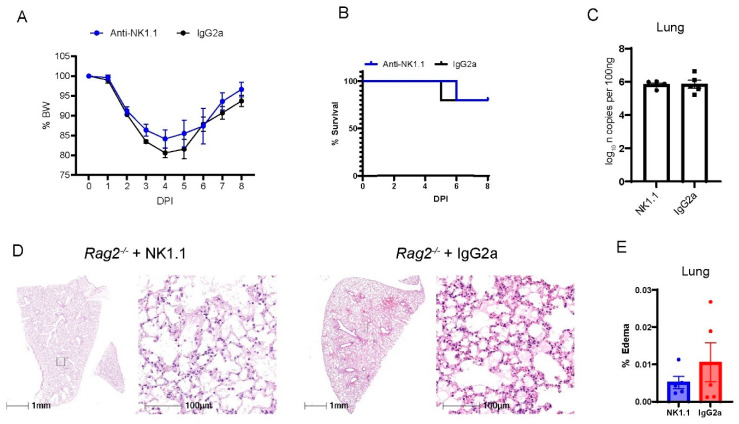
The depletion of NK cells in *Rag2*^−/−^ mice did not impact MA30-induced diseases. Eighteen-week-old male *Rag2*^−/−^ mice were infected with medium dose of MA30 (2 × 10^4^ TCID_50_) and treated with either anti-NK or the isotype control (*n* = 5/group). (**A**) Body weight (BW) changes in the mice. The infected mice were monitored for 8 DPI and no difference was found using two-way ANOVA. (**B**) Survival rate in the mice. No difference was found in the mice using log-rank (Mantel-Cox) test. (**C**) Viral subgenomic N RNA analysis. No difference was found in viral subgenomic N RNA of the mice. (**D**) Lung histological changes. Representative images show comparable edema between anti-NK1.1 treated and control groups. (**E**) Quantitative analysis of edema. There is no significant difference found between them using unpaired *t*-test.

## Data Availability

Data are contained within the article and Appendix A.

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
