# Peer review of "Natural Killer Cells Do Not Attenuate a Mouse-Adapted SARS-CoV-2-Induced Disease in Rag2−/− Mice"

_viruses, 2024, doi:10.3390/v16040611_

Round 1

Reviewer 1 Report

Comments and Suggestions for Authors

Here Ellsworth et al from the Qin lab use MA30 SARS-CoV-2 to study and compare the disease progression in B6, Rag2-/- and K18-hACE2 mice. They find differences in extent of infected cell types and location of infection (Alveolar epithelial cells: K18-hACE2 versus Bronchial epithelial cells: B6). They then use Rag2-/- mice to study the contribution of B/T cells and NK cells during MA30 infection and find that B/T cells contribute to alleviating disease whereas NK cells do not. I have following comments for the authors to consider:

Major:

1.   T and/or B cells but not Natural Killer cells protect against acute severe COVID-19 in mice”. The title is an exaggeration (protection). The authors show that T and B cells may contribute to diminishing MA30-induced disease using RAG2-/- mice.

2.  Do the authors expect a large contribution of B and T cells during acute phase of infection? Activation and expansion of both T cells and B cells takes time 7-14 days. So, the data are not surprising. A better experiment would have been to immunodeplete CD4+ T cells, CD8+ T cells and B cells during acute phase individually to make the point. While the morbidity increases significantly in absence of T/B cells, importantly, Rag2-/- mice recover from disease rather than succumbing to infection. Hence the use of the word “protection” is not accurate. It should be “contributes.”

3.  NK cells are innate cells that can directly kill target cells or can do so via binding to Fc regions of bound antibodies. The NK cell depletion experiment suggest that the innate killing function of NK may not contribute to alleviate MA30-induced disease. This is despite hyperactivation of NK cells in Rag2-/- mice (10.1038/s42003-023-05606-3). That said, one must exercise caution while interpreting these data. Ideally NK cell depletion in B6 mice would have been ideal experiment for better physiological relevance. But one does not expect the outcome to change dramatically.

4. The authors do not use cell type specific markers to discern if the infected cells are indeed alveolar epithelial cells. They overly rely on location and shape of the cells. Using markers like Spc (AT2)/podoplanin (AT1)/Epcam (BECs) can allow the authors to better characterize the infected cell types by FACS or by imaging and substantiate their claim. Quality of images are also poor and a higher-resolution images can help.

5. Inflammation is monitored primarily with histopathological analyses. Evaluation of inflammatory cytokines or mRNA expression can help to corroborate the data.

6. “Although our study corroborates data that suggest NK cells are exhausted and thereby play a negligible role in the acute phase of severe COVID-19 disease in patients” I did not see any data showing NK cells were exhausted upon MA30 infection. Please rewrite and exercise caution before making claim(s) that are not substantiated with data. 

7. Would the author expect to see similar differences in infection profile if MA30 was used in K18-hACE2 mice to compare with WA1 infection?

Minor:

1. There are several instances of SARS-CoV2 instead of SARS-CoV-2 (hyphen is missing)

2. “Comparison of MA30 vs the K18-ACE SARS-CoV2-WA1 model”: ACE2 and not ACE

3.     “extends the findings of Perlman”. I urge the authors to use proper scientific language in written manuscripts. 

4.  “MA30 model's value”: The value of MA30 model.

5. The authors use COVID-19 to describe MA30-induced disease: "Deficiency of Rag2 accelerates COVID-19 in mice". Maybe the authors should use COVID-19 to indicate human disease. Humans do not experience dramatic weight loss like mice during COVID-19. Mouse physiology is different, and we use them as models.

Comments on the Quality of English Language

Overall quality is good but there are signs of lazy writing in some instance.

Author Response

One by one responses to reviewers’ comments:

Major comments from reviewer 1.

Comment 1.   “T and/or B cells but not Natural Killer cells protect against acute severe COVID-19 in mice”. The title is an exaggeration (protection). The authors show that T and B cells may contribute to diminishing MA30-induced disease using RAG2-/- mice.

Response: We agree with your comment. To reflect this point, we have changed the title as follows: “T and/or B cells but not Natural Killer cells attenuate a mouse-adapted SARS-CoV-2-induced disease.”

Comment 2.  Do the authors expect a large contribution of B and T cells during acute phase of infection? Activation and expansion of both T cells and B cells takes time 7-14 days. So, the data are not surprising. A better experiment would have been to immunodeplete CD4+ T cells, CD8+ T cells and B cells during acute phase individually to make the point. While the morbidity increases significantly in absence of T/B cells, importantly, Rag2-/- mice recover from disease rather than succumbing to infection. Hence the use of the word “protection” is not accurate. It should be “contributes.”

Response:  We agree with you. The relative contributions of CD4+ T cells, CD8+ T cells or B cells to the attenuation of MA30-induced COVID phenotype during the acute phase remain elusive and require further investigation with the elimination of these cells in the mice by using cell-targeted ablation tools. We have included these points in the discussion.  In addition, we agree with the reviewer's comments about the word “protection” in this context throughout the content and have changed the title and modified the document by replacing the word “protect” with attenuate or mitigate accordingly.

Comment 3.  NK cells are innate cells that can directly kill target cells or can do so via binding to Fc regions of bound antibodies. The NK cell depletion experiment suggests that the innate killing function of NK may not contribute to alleviating MA30-induced disease. This is despite the hyperactivation of NK cells in Rag2-/- mice (10.1038/s42003-023-05606-3). That said, one must exercise caution while interpreting these data. Ideally NK cell depletion in B6 mice would have been ideal experiment for better physiological relevance. But one does not expect the outcome to change dramatically.

Response: We acknowledge that depletion of NK cells in B6 mice would be an ideal experiment for the physiological relevance of NK cells, as NK cells may crosstalk with B/T cells during the infection. To exercise caution in interpreting our results, we have added to the discussion to highlight this point: “We found that the absence of these cells leads to prolonged viral persistence and incomplete weight recovery. However, the relative contributions of CD4+ T cells, CD8+ T cells or B cells to the attenuation of MA30-induced COVID phenotype during the acute phase remain elusive and require further investigation with the elimination of these cells in the mice by using cell-targeted ablation tools (42-46). Additionally, we investigated the role of natural killer cells in severe COVID-19 using Rag2-/- mice, a mouse model deficient in T and B maturation. Our finding suggests that the innate function of NK may not contribute to alleviating MA30-induced disease. Of note, a previous study has indicated that there was hyperactivation of NK cells in Rag2-/- mice(47). Although NK cell depletion in B6 mice would provide further insight into the role of NK in COVID-19, we do not expect the outcome to change dramatically considering that this hyperactivation of NK cells in Rag2-/- mice would have been suppressed by our cell ablation approach.”

Comment 4. The authors do not use cell type specific markers to discern if the infected cells are indeed alveolar epithelial cells. They overly rely on location and shape of the cells. Using markers like Spc (AT2)/podoplanin (AT1)/Epcam (BECs) can allow the authors to better characterize the infected cell types by FACS or by imaging and substantiate their claim. Quality of images are also poor and a higher-resolution images can help.

Response: We stained the marker Spc (AT2)/podoplanin to better characterize the infected cell types and taken higher quality images of the infected lung. We found that the infected cells in the K18 mice but not the B6 mice are mainly Spc (AT2) positive cells, which is included in Supplemental Figure 2. The finding in the infected K18 is consistent with the previous observation published in PMID: 34668775. We have taken higher-resolution and color-enhanced images for the Figure 3D and Supplemental figure 3.

Comment 5. Inflammation is monitored primarily with histopathological analyses. Evaluation of inflammatory cytokines or mRNA expression can help to corroborate the data.

Response: We fully agree with your comment.  To address the concern, we have conducted a flow cytometry analysis of T, B, and NK cells in the lung and circulation of the infected mice (Supplemental figures 5 and 6).  We have also conducted bulk RNA seq analysis of the infected lungs of the mice.  We have included transcriptomic changes in Figure 4A and 4B and supplemental tables 1 and 2.  Certainly, these additional results enhance the quality of our paper and corroborate the data better.  We have expanded these in result and discussion section accordingly.

Comment 6. “Although our study corroborates data that suggest NK cells are exhausted and thereby play a negligible role in the acute phase of severe COVID-19 disease in patients” I did not see any data showing NK cells were exhausted upon MA30 infection. Please rewrite and exercise caution before making claim(s) that are not substantiated with data. 

Response: Thank you so much for the great comment.  To address this concern, we have modified the wording in the document to exercise caution. The new sentence now reads as follows in the discussion: “NK cells are exhausted and thereby play a negligible role in the acute phase of severe COVID-19 disease in patients. Our results support those clinical findings though further investigation of the exhausted status of NK cells is required.”

Comment 7. Would the author expect to see similar differences in infection profile if MA30 was used in K18-hACE2 mice to compare with WA1 infection?

Response: We do not know for sure. It would be interesting to see MA30-infected K18 mice develop a more severe (or similar) COVID-19 phenotype as compared with WA1-infected K18 mice, and whether the viral tropism of MA30-infected K18 mice significantly changes from the viral tropism of MA30-infected B6 mice.

Minor comments from Reviewer 1:

Comment 1. There are several instances of SARS-CoV2 instead of SARS-CoV-2 (hyphen is missing)

Response:  We have fixed them.

Comment 2. “Comparison of MA30 vs the K18-ACE SARS-CoV2-WA1 model”: ACE2 and not ACE

Response:  We have fixed them.

Comment 3. “extends the findings of Perlman”. I urge the authors to use proper scientific language in written manuscripts. 

Response: Yes, you are right.  We have modified our statement as follows: “Notably, our study further demonstrates that aging mice from only 8 to 12 weeks significantly exacerbates these symptoms, highlighting the age-dependent severity of the disease.

Comment 4.  “MA30 model's value”: The value of MA30 model.

Response:  We have modified it accordingly.

Comment 5. The authors use COVID-19 to describe MA30-induced disease: "Deficiency of Rag2 accelerates COVID-19 in mice". Maybe the authors should use COVID-19 to indicate human disease. Humans do not experience dramatic weight loss like mice during COVID-19. Mouse physiology is different, and we use them as models.

Response:  Correct.  To address this concern, we have replaced COVID-19 in mice with MA30-induced disease entirely. 

Comment 6. Overall quality is good but there are signs of lazy writing in some instances.

Response:  We have cleaned up the errors in the paper.

Reviewer 2 Report

Comments and Suggestions for Authors

In the present manuscript, Dr Ellsworth report on the roles of B, T and NK cells on SARS-CoV2 infection in a mouse model of COVID-19.

The authors inoculated B6 mice with various doses of a mouse-adapted SARS-CoV2 virus and evaluated different parameters during the follow up time. The authors also used an original SARS-CoV2 virus to infect a “humanized” mouse model bearing the human ACE2 receptor.

The authors observed weight lost, high viral load in the lungs, link of survival upon infection with the quantity of virus inoculum, massive infection of lung epithelial cells and minimal role of natural killer cells. The authors concluded that B and/or T cells protect against acute severe COVID-19, while the role of NK cells remains marginal.

This is an important and welcome work performed by the authors. The manuscript is clearly written and the results well reported. The authors are commended for that.

The work presents some limitations.

While the title of the manuscript is affirmative, it does not present mechanistic data to aid explain their findings:

-       why did the authors not quantify cytokines profile changes during the course of infection?

-       Why did the authors not quantify antibodies induced by infection over time?

-       Why did they not evaluate cell-mediated effector activity of B, T and NK cells?

-        What about neutralizing activity of antibodies produced during infected by mice in the two different compared to RAG2 deficient mice?

This is unfortunate because all these parameters are obtained are standard experiments that could have been done with biological material available during the follow up.

Hence, the reviewer is left with a taste of an incomplete, descriptive work.

A minor comment is the absence of page numbers and line numbers as per-journal recommendations.

Author Response

Comments from reviewer 2.

Comment 1: While the title of the manuscript is affirmative, it does not present mechanistic data to aid explain their findings:

Response: Based on review 1 and your suggestion, we have changed the title to more accurately reflect the results of the paper: “T and/or B cells but not Natural Killer cells attenuate a mouse-adapted SARS-CoV-2-induced disease.”

Comment 2: why did the authors not quantify cytokines profile changes during the course of infection?

Response: We did not collect serum and the blood we collected for Flow analysis of the T/B cells and NK cells. We include this data in supplemental figures 5 and 6.  We have performed bulk RNA seq of the infected lungs and included the gene and pathway changes in the infected Rag2-/- as compared to B6 mice as we have described above.

Comment 3: Why did the authors not quantify antibodies induced by infection over time?

Response: We did not collect serum and so no antibody could be detected. Our additional data on bulk RNA analyses reveals that there is significant dysregulation of antibody production pathways such as T, B cell receptor signaling and downregulated key genes related to these pathways.  We have expanded our discussion as follows:” Further, the infected Rag2-/- mice had a significant dysregulation in the primary immunodeficiency, B cell receptor signaling pathway, T cell receptor signaling pathway, Th17 cell differentiation, and Th1 and Th2 cell differentiation.  These pathways are closely related to T and B cell differentiation and activation, and antibody production. Key genes related to these pathways (for example Cd8a for primary immunodeficiency and antibody production, Cd22, Cd79b and Igj for B cell receptor signaling pathway, Cd8a for T cell receptor signaling pathway and H2-Ob for Th17 cell differentiation, and Btla and H2-Ob for Th1 and Th2 cell differentiation and activation) were down-regulated in the infected Rag2 deficient mice. These defects and gene down-regulation can be attributed to deficiency of Rag2 function in Rag2-/- mice in the response to MA30 infection. However, how the down-regulated genes and dysregulated pathways attenuate the acute infection remains unclear”.

Comment 4: Why did they not evaluate cell-mediated effector activity of B, T and NK cells?

Response: We now included the flow data measurements for B, T, NK cells in circulation and the lung in Supplemental Figures 5 and 6. However, the effector activity requires additional experiments. We have added this limitation to the discussion of the paper as follows: “Our results support those clinical findings though further investigation of the exhausted and activated status of NK cells is required.”

Comment 5: What about the neutralizing activity of antibodies produced during infection by mice in the two different compared to RAG2 deficient mice?

Response: We did not detect the antibody level; however, we do see T/B cell expansion in the B6 mice but not in the Rag2 mice, such that the antibody titer is assumed to be none or less.

Comment 6: This is unfortunate because all these parameters are obtained are standard experiments that could have been done with biological material available during the follow up. Hence, the reviewer is left with a taste of an incomplete, descriptive work.

Response: We did not collect the blood from these mice except for flow cytometry analysis, which we now include in the paper. We also measured the bulk mRNA in the lungs in this revision.

Comment 7: A minor comment is the absence of page numbers and line numbers as per-journal recommendations.

Response: We have added page numbers and line numbers in the revision.

Reviewer 3 Report

Comments and Suggestions for Authors

The manuscript by Ellsworth and colleagues describes the involvement of T and/or B cells against acute SARS-CoV-2 infection of mice. The authors examine the pathogenesis of the MA30 strain of SARS-CoV-2 infection in C57BL/6 mice (B6) and the primary WA1 strain of SARS-CoV-2 in K18 mice (transgenic for hACE2). While the MA30 strain in B6 mice and the WA-1 strain in K18 mice are both pathogenic in their respective strains, the infection appears to be somewhat different. The salient findings of the study are that a) inoculation of B6 mice with MA30 results in infection of the nasal cavity and epithelial cells along the respiratory tract but little in the alveoli while WA1 infection of K18 mice resulted in infection of the nasal cavity and the alveoli. The observation of the little infection of the alveoli by the MA30 strain is likely due to the absence of the human ACE2 receptor on type 2 alveolar epithelial cells.

Comments:

Comment #1: As the manuscript attempts to discern differences in the two models as they pertain to SARS-CoV-2 infection of humans, a better comparison of the histopathology observed in severely infected humans versus the two mouse models is warranted.

Comment #2: The authors present data that if you inoculate Rag-/- mice (which are defective for the development of T and B cells), the disease caused by MA30 is more severe. This is not surprising as the elimination of two arms of the adaptive immune response against viral pathogens, the observed pathogenesis will be predictably more severe.

Comment #3: The role of NK cells in the clearance of SARS-CoV-2 has been studied in humans and it is generally thought that NK cells probably do not play a major role in the clearance of SARS-CoV-2. It is known that a severe outcome in SARS-CoV-2 infection of humans is associated with decreased numbers of NK cells in peripheral blood and concomitant NK cell hyperactivation, leading to distinct immunotypes related to disease severity. The authors should examine NK cell numbers and hyperactivation in B6 mice inoculated with MA30 and determine if NK cell numbers decrease and undergo hyperactivation in the K18 mice inoculated with the MA-1 strain of SARS-Cov-2. Finally, the authors also need to discuss their results in these mice as they pertain to SARS-CoV-2 infection of humans.

Comment #4: As the manuscript attempts to discern differences in the two models as they pertain to SARS-CoV-2 infection of humans, a better comparison of the histopathology observed in severely infected humans versus the two mouse models is warranted.

Comment #5: Measurement of sub-genomic N RNA is not equivalent to viral load. It quantifies the number of sub-genomic N RNA molecules that are present. For example, when viral loads are measured in HIV-1-infected patients, viral RNA is measured from the virus present in the circulation. This needs to be clarified.

Comment #6: In the legend of Figure 2, the sixth line begins with MA20. It should be corrected to MA30.

Comment #7: In Figure 3D, it is extremely hard to see the immunofluorescence staining of the spike protein. The authors should consider either taking micrographs with a longer exposure or repeating the experiment.

Comment #8: TCID50 is generally written as TCID50. Please correct.

Comment #9: (throughout the manuscript) T-test should be written as t-test. Please correct.

Comment #10: It is not stated if the Rag-/- mice are B6-derived. Please clarify.

Supplemental Figure 1A and B: It is unclear what is being shown. The legend does not state what the red arrowheads are pointing to. Also, the time at which these mice were sacrificed should be given. Please clarify.

Supplemental Figure 2: In this Figure, the authors show the lungs of five Rag-/- and B6 mice immunostained for Spike protein to show the differences in severity of infection. However, except for the panel of the second Rag-/-  mouse, the remaining four Rag-/- mice appear to only have background staining. Also, there was little to no staining in B6 lungs inoculated with MA30. The authors need to provide information on inoculum size and the time after inoculation that the mice were sacrificed.

Supplemental Figure 3: The authors do not state what virus was used, the inoculum size, and the time after inoculation that the mice were sacrificed to inoculate these mice. I assume it is MA30.

Comments on the Quality of English Language

No comments, only minor corrections.

Author Response

Comments from the reviewer 3.

Comment 1: The salient findings of the study are that a) inoculation of B6 mice with MA30 results in infection of the nasal cavity and epithelial cells along the respiratory tract but little in the alveoli while WA1 infection of K18 mice resulted in infection of the nasal cavity and the alveoli. The observation of the little infection of the alveoli by the MA30 strain is likely due to the absence of the human ACE2 receptor on type 2 alveolar epithelial cells.

Response: Thank you so much for your great comments.  Yes, indeed, we have demonstrated that WA1 infection of K18 mice results in hACE2 receptor on type 2 cells as shown in Supplemental Figure 2.

Comment #2: As the manuscript attempts to discern differences in the two models as they pertain to SARS-CoV-2 infection of humans, a better comparison of the histopathology observed in severely infected humans versus the two mouse models is warranted.

Response: We fully agree with your comments.  To address your suggestion, we expand our discussion as follows: “Comparing the MA30-infected B6 mice with traditional models using wild-type viral strains and humanized K18 mice, we observed distinct pathologic phenotypes. The infected B6 mice exhibited epithelial cell infections in the entire airway––including the nasal cavity, bronchus and bronchioles and alveoli––while the infected K18 mice exhibited epithelial cell infections in the nasal cavity and alveoli, with little to no infection in the bronchus and bronchioles. This viral tropism is comparable to the viral tropism observed from autopsy studies of deceased COVID patients shown in the previous studies.  The infected-B6 mice developed more severe peri-bronchial inflammation with similar pulmonary edema and less lung interstitial inflammation compared with the infected-K18 mice. The pulmonary alveolar edema is one of major pathological changes seen in severe COVID-19.  Together, the viral tropism and pathological changes seen in the infected B6 mice recapitulate several key histopathological features seen in severe COVID-19 patients. This underscores the value of MA30 model in understanding the pathogenesis of severe COVID-19, especially its pulmonary complications.”

Comment #3: The authors present data that if you inoculate Rag2-/- mice (which are defective for the development of T and B cells), the disease caused by MA30 is more severe. This is not surprising as the elimination of two arms of the adaptive immune response against viral pathogens, the observed pathogenesis will be predictably more severe.

Response: We agree with your comment.

Comment #4: The role of NK cells in the clearance of SARS-CoV-2 has been studied in humans and it is generally thought that NK cells probably do not play a major role in the clearance of SARS-CoV-2. It is known that a severe outcome in SARS-CoV-2 infection of humans is associated with decreased numbers of NK cells in peripheral blood and concomitant NK cell hyperactivation, leading to distinct immunotypes related to disease severity. The authors should examine NK cell numbers and hyperactivation in B6 mice inoculated with MA30 and determine if NK cell numbers decrease and undergo hyperactivation in the K18 mice inoculated with the MA-1 strain of SARS-Cov-2. Finally, the authors also need to discuss their results in these mice as they pertain to SARS-CoV-2 infection of humans.

Response: To address this question, we have quantified the number of NK cells at 3 DPI. (NK1.1 stain.T cell B markers, monocytes) shown in Supplemental figures 5 and 6.  We have added the results in the result section as follows: “Previous studies have shown that the patients at the early phase had increased percentages of NK cells in the circulation (35) and in BAL fluid (36) as compared with the patients before the infection.  Consistently, we have found that MA30-infected B6 and Rag2-/- mice had a significantly higher percentage of NK cells in blood and lung than naïve B6 mice (Supplemental Figure 6)”.

Comment #5: Measurement of sub-genomic N RNA is not equivalent to viral load. It quantifies the number of sub-genomic N RNA molecules that are present. For example, when viral loads are measured in HIV-1-infected patients, viral RNA is measured from the virus present in the circulation. This needs to be clarified.

Response: We changed the the viral load to subgenomic N RNA throughout papers and figures.

Comment #6: In the legend of Figure 2, the sixth line begins with MA20. It should be corrected to MA30.

Response:  Fixed

Comment #7: In Figure 3D, it is extremely hard to see the immunofluorescence staining of the spike protein. The authors should consider either taking micrographs with a longer exposure or repeating the experiment.

Response: We enlarge image.

Comment #8: TCID50 is generally written as TCID50. Please correct.

Response: fixed

Comment #9: (throughout the manuscript) T-test should be written as t-test. Please correct.

Response: fixed

Comment #10: It is not stated if the Rag-/- mice are B6-derived. Please clarify.

Response: Rag2-/- mice are in a C57BL/6J (B6) background (Stock no: 008449).  We have specified this in the paper.

Comment 11. Supplemental Figure 1A and B: It is unclear what is being shown. The legend does not state what the red arrowheads are pointing to. Also, the time at which these mice were sacrificed should be given. Please clarify.

Response: We have fixed all in the legend by specifying what the red arrowheads point out and adding the days post infection for sacrificing mice.

Comment 12: Supplemental Figure 2: In this Figure, the authors show the lungs of five Rag-/- and B6 mice immunostained for Spike protein to show the differences in severity of infection. However, except for the panel of the second Rag-/- mouse, the remaining four Rag-/- mice appear to only have background staining. Also, there was little to no staining in B6 lungs inoculated with MA30. The authors need to provide information on inoculum size and the time after inoculation that the mice were sacrificed.

Response: We have enlarged and magnified the image for easier viewing. And we clarify the details of the inoculum.

Supplemental Figure 3: The authors do not state what virus was used, the inoculum size, and the time after inoculation that the mice were sacrificed to inoculate these mice. I assume it is MA30.

Response:  We have clarified these in the figure legend.  

Round 2

Reviewer 1 Report

Comments and Suggestions for Authors

The authors have tried to earnestly improve the manuscript based on reviewer comments. However issues remain that need to be addressed in my opinion. There are overinterpretations and the study lacks correct controls.

1. Since the authors did not investigate T and B cell function in detail (second reviewers comments), it is better to tone down the title and limit interpretation to only NK cells. The severity seen in Rag-/- mice is overinterpreted as many pathways can be non-functions in this immunodeficient background. Lack of B and T cell depletion experiments in immunocompetent B6 mice adds to the complication in interpretation. I would suggest the authors to be careful with their interpretations.

2. There are several instances where authors have continued to use colloquial terminology.: "K18 mice" instead of specific terminology "K18-hACE2 mice". CoV2 instead of SARS-CoV-2 (Figure 2 title) Please correct.

3. Typos still exist: eg: 5 X 104 TCID50

4. Supplemental 2: the description and conclusion is not correct. The authors just use Spc (abbreviated incorrectly in the title legend) to mark AT2 cells when in K18-hACE2 mice there is infection of AT1 cells (pdpn ; not done) cells as well (in addition to AT2). This is one of the contributing reason (expansion of host tropism) for severity seen in K18-hACE2 mice. Please correct.

5. Supplemental 5: The interpretation is incorrect. The authors are comparing naive B6 mice with infected Rag 2-/- mice. The control for infected Rag2-/- mice should be naive Rag2-/- mice.  This applies for Sup 6 figure as well. Second I am unable to understand the point of this experiment in response to "inflammation" comment. The author should have ideally estimated neutrophils and Ly6C hi monocytes to measure inflammation induced infiltration of inflammatory cells in addition to NK cells that the authors have done. Third it should be ideally done for lungs (done) and BAL fluids.

6. All the authors had to do was to carryout a simple RNA based real-time PCR analysis of marker inflammatory cytokine mRNAs in the lung tissue. The authors opted to carry out RNAseq analyses instead. Also the comparison should be between naive B6 and infected B6; naive Rag2-/- vs infected Rag 2-/- mice. Such a comparison then opens up the possibility for  the authors to compare infected B6 with infected Rag2-/- mice and comment on the level of inflammation. The lack of appropriate control results in interpretations like "Cxcl9, Cxcl1. Ccl2, Ccl7, Cxcl2 and Cxcl5 were significantly up-regulated in lungs of the infected Rag2-/- mice as compared the infected B6 mice" that can be completely misleading. I urge the authors to be careful. RNAseq would have been better applied to test if SARS-CoV-2 infection results in dysfunctional NK cells by purifying the NK cells.

Comments on the Quality of English Language

Some typos are still present, which need to be fixed

Author Response

Comments from Review 1:

Comment 1: Since the authors did not investigate T and B cell function in detail (second reviewers comments), it is better to tone down the title and limit interpretation to only NK cells. The severity seen in Rag-/- mice is overinterpreted as many pathways can be non-functions in this immunodeficient background. Lack of B and T cell depletion experiments in immunocompetent B6 mice adds to the complication in interpretation. I would suggest the authors to be careful with their interpretations.

Response: The new title has been changed into: “Natural killer cells do not attenuate a mouse-adapted SARS-CoV-2-induced disease in Rag2-/- mice.

Comment 2: There are several instances where authors have continued to use colloquial terminology.: "K18 mice" instead of specific terminology "K18-hACE2 mice". CoV2 instead of SARS-CoV-2 (Figure 2 title) Please correct.

Response:  All are fixed in this revision.

Comment 3: Typos still exist: eg: 5 X 104 TCID50

Response:  We have corrected them.

Comment 4: Supplemental 2: the description and conclusion is not correct. The authors just use Spc (abbreviated incorrectly in the title legend) to mark AT2 cells when in K18-hACE2 mice there is infection of AT1 cells (pdpn ; not done) cells as well (in addition to AT2). This is one of the contributing reason (expansion of host tropism) for severity seen in K18-hACE2 mice. Please correct.

Response:  Thanks.  To address this specific question, we have revised our description in the results and discussion sections as the follows:

In the result section, we have described as the following: “We found that many alveolar type 2 (AT2) cells were infected in the K18-hACE2 mice as compared with the B6 mice.”

In the discussion section, we have described as the following: “Of note, the previous study also documented that in the infected K18-hACE2 mice, there is an infection of alveolar type I cells (AT1) cells in addition of AT2 infection, which is one of the contributing factor for severity seen in the infected K18-hACE2 mice (44). However, whether MA30 also infects the AT1 cells in B6 mice remains unclear and requires further investigation.”

Comment 5:  Supplemental 5: The interpretation is incorrect. The authors are comparing naive B6 mice with infected Rag 2-/- mice. The control for infected Rag2-/- mice should be naive Rag2-/- mice.  This applies for Sup 6 figure as well. Second I am unable to understand the point of this experiment in response to "inflammation" comment. The author should have ideally estimated neutrophils and Ly6C hi monocytes to measure inflammation induced infiltration of inflammatory cells in addition to NK cells that the authors have done. Third it should be ideally done for lungs (done) and BAL fluids.

Response: We fully agree with the comment on comparing Naïve B6 mice with infected Rag2-/- mice.  To address this comment, in Supplemental Figure 5, we have moved the data obtained from naïve B6 mice.  We only compare the infected B6 with Rag2-/- mice this figure.  In the Supplemental Figure 6, we only compare the NK cell changes in naïve B6 with the infected B6 mice to show the NK cells in response to the infection, which is requested by the previous reviewer 3. In supplemental figure 6, we also compare the NK cell changes in the infected B6 with Rag2-/- mice.

The comments on monitoring the inflammation by determining neutrophils and Ly6C hi monocytes in the BAL and lungs of the infected mice. This is a great comment.  Unfortunately, in our previous experiments, we did not collect BAL and stain the specific marker for these cells in the blood and lung samples.  Therefore, we cannot provide those data. 

Comment 6: All the authors had to do was to carryout a simple RNA based real-time PCR analysis of marker inflammatory cytokine mRNAs in the lung tissue. The authors opted to carry out RNAseq analyses instead. Also the comparison should be between naive B6 and infected B6; naive Rag2-/- vs infected Rag 2-/- mice. Such a comparison then opens up the possibility for  the authors to compare infected B6 with infected Rag2-/- mice and comment on the level of inflammation. The lack of appropriate control results in interpretations like "Cxcl9, Cxcl1. Ccl2, Ccl7, Cxcl2 and Cxcl5 were significantly up-regulated in lungs of the infected Rag2-/- mice as compared the infected B6 mice" that can be completely misleading. I urge the authors to be careful. RNAseq would have been better applied to test if SARS-CoV-2 infection results in dysfunctional NK cells by purifying the NK cells.

Response:  Based on this comment, we have removed the statement: “Cxcl9, Cxcl1. Ccl2, Ccl7, Cxcl2 and Cxcl5 were significantly up-regulated in lungs of the infected Rag2-/- mice as compared the infected B6 mice” in the discussion section. 

Reviewer 2 Report

Comments and Suggestions for Authors

I would like to congralute the authors for having considered all my queries, including the minor ones, in the revised version of their report. I have no additional remarks

Author Response

There is no comment to be addressed

Reviewer 3 Report

Comments and Suggestions for Authors

In the revised manuscript by Ellsworth and colleagues, they investigate the role of NK cells in B6 mice and Rag2-/- mice following inoculation with the MA30 strain of SARS-CoV-2.  I have reviewed the manuscript and the authors have addressed all my previous comments. I just have few minor comments:

Line 118: "veterinarian" should be "veterinary"

Line 206: "2×10^5" should be "2×105"

Lines 234-235: "P < 0.0001" and "P < 0.05" should be "p< 0.0001 and p< 0.05" The use of a capitalized or lowercase p varies with the journal. I have used lowercase and italicized.  I don't know this journal's convention but just be consistent.

Line 375: "t-test" should be "t-test"

Comments on the Quality of English Language

No problem

Author Response

Comments from reviewer 3:

Comment 1: Line 118: "veterinarian" should be "veterinary"

Response:  We corrected it already.

Comment 2: Line 206: "2×10^5" should be "2×105"

Response: We have revised it accordingly.

Comment 3: Lines 234-235: "P < 0.0001" and "P < 0.05" should be "p< 0.0001 and p< 0.05" The use of a capitalized or lowercase p varies with the journal. I have used lowercase and italicized.  I don't know this journal's convention but just be consistent.

Response: Line 375: "t-test" should be "t-test"